# MiRNAs and mRNAs Analysis during Abdominal Preadipocyte Differentiation in Chickens

**DOI:** 10.3390/ani10030468

**Published:** 2020-03-11

**Authors:** Xiangfei Ma, Junwei Sun, Shuaipeng Zhu, Zhenwei Du, Donghua Li, Wenting Li, Zhuanjian Li, Yadong Tian, Xiangtao Kang, Guirong Sun

**Affiliations:** College of Animal Science and Veterinary Medicine, Henan Agricultural University, Zhengzhou 450002, China; mxf1228@126.com (X.M.); jw2013au@126.com (J.S.); zhu15039087513@126.com (S.Z.); x13592547117@126.com (Z.D.); lidonghua6656@126.com (D.L.); liwenting@henau.edu.cn(W.L.); lizhuanjian@163.com (Z.L.); ydtian111@163.com (Y.T.); xtkang2001@263.net (X.K.)

**Keywords:** RNA sequencing, abdominal fat cells, abdominal fat deposition

## Abstract

**Simple Summary:**

We sequenced the miRNAs and mRNAs of preabdominal fat cells and differentiated adipocytes, and target genes of miRNA combined with mRNA transcriptome data jointly. We found that the MAPK signal pathway, insulin signal pathway, fatty acid metabolism, ECM( extracellular matrix)–receptor interaction, and other signal pathways were involved in the differentiation of preabdominal fat cells. In addition, we found that some miRNAs–mRNAs combinations were strongly related to the differentiation of fat cells (miR-214−*ACSBG2*, *NFKB2*, *CAMK2A*, *ACLY*, *CCND3*, *PLK3*, *ITGB2*; miR-148a-5p−*ROCK2*; miR-10a-5p−*ELOVL5*; miR-146b-5p−*LAMA4*; miR-6615-5p−*FLNB*; miR-1774−*COL6A1*). Our findings provide important resources for the study of adipocyte differentiation.

**Abstract:**

The excessive deposition of abdominal fat has become an important factor in restricting the production efficiency of chickens, so reducing abdominal fat deposition is important for improving growth rate. It has been proven that miRNAs play an important role in regulating many physiological processes of organisms. In this study, we constructed a model of adipogenesis by isolating preadipocytes (Ab-Pre) derived from abdominal adipose tissue and differentiated adipocytes (Ab-Ad) in vitro. Deep sequencing of miRNAs and mRNAs expressed in Ab-Pre and Ab-Ad groups was conducted to explore the effect of miRNAs and mRNAs on fat deposition. We identified 80 differentially expressed miRNAs (DEMs) candidates, 58 of which were up-regulated and 22 down-regulated. Furthermore, six miRNAs and six mRNAs were verified by qRT-PCR, and the results showed that the expression of the DEMs and differentially expressed genes (DEGs) in the two groups was consistent with our sequencing results. When target genes of miRNA were combined with mRNA transcriptome data, a total of 891 intersection genes were obtained, we predicted the signal pathways of cross genes enrichment to the MAPK signal pathway, insulin signal pathway, fatty acid metabolism, and ECM–receptor interaction. Meanwhile, we constructed miRNA and negatively correlated mRNA target networks, including 12 miRNA–mRNAs pairs, which showed a strong association with the abdominal adipocyte differentiation (miR-214−*ACSBG2*, *NFKB2*, *CAMK2A*, *ACLY*, *CCND3*, *PLK3*, *ITGB2*; miR-148a-5p−*ROCK2*; miR-10a-5p−*ELOVL5*; miR-146b-5p−*LAMA4*; miR-6615-5p−*FLNB*; miR-1774−*COL6A1*). Overall, these findings provide a background for further research on lipid metabolism. Thus, we can better understand the molecular genetic mechanism of chicken abdominal fat deposition.

## 1. Introduction

With the development of intensive feeding, the daily growth rate and bodyweight of chickens were tended to be selected, but the accumulation of high abdominal fat is still an obstacle [1,2]. In poultry, excessive abdominal fat deposition not only increases the cost of feeding and negatively affects the appearance of products, but also poses a threat to the health of consumers [3,4]. Our previous studies have shown that abdominal preadipocytes have higher adipogenic differentiation ability than intramuscular preadipocytes [5], so reducing abdominal fat deposition is of great significance to the poultry industry. Abdominal fat deposition is affected by heredity, epigenetics, and hormones. However, at present, the research on abdominal fat deposition in chicken is not systematic and deep enough, and the molecular genetic mechanism of the formation of such complex traits has not been found out yet.

MiRNAs are a kind of non-coding single-stranded RNA molecule, which are encoded by endogenous genes and are 21–24 nucleotides in length. miRNAs have the function of regulating gene expression at translation level or transcriptional level. Recognition of target mRNA of miRNAs is mainly achieved through the interaction between seed sequences and target gene 3′UTR’s miRNAs regulatory element (MRE). The regulation of mature miRNAs on target genes depends on the complementarity between the seed sequence of miRNAs and the 3′UTR regulatory region of the target gene, leading to translation inhibition or gene splicing [6,7,8]. In recent years, many studies have found that miRNAs are involved in many biological processes, such as disease and metabolism [9,10]. Some studies have also shown that miRNAs play an important role in regulating adipocyte differentiation and lipid metabolism [11,12]. *KLF13* targeted by miR-425-5p inhibits adipocyte differentiation during porcine intramuscular adipocyte differentiation [13] and miR-26a promoting 3T3-L1 adipocyte differentiation via targeting regulatory *PTEN* [14]. Regulation of miRNAs on human adipocyte differentiation has also been reported. miR-361-5p and miR-574-5p may be associated with hypertrophy of human white adipose tissue (WAT) by affecting the expression of *EBF1* [15]. However, there are few studies on miRNA of chicken abdominal adipocyte.

Gushi chicken is an excellent breed of laying hens and meat production in Gushi county, Henan province, China. Although Gushi chicken has many favorable characteristics, the accumulation of abdominal fat is still a problem that needs to be solved. It was reported that the heritability of abdominal fat (0.82) was significantly higher than that of live weight (0.55), breast muscle (0.55), leg (0.51), and thigh (0.31) at the time of selection [16]. The molecular mechanisms related to abdominal fat regulation remain unclear in chicken. In order to reveal the crucial molecular mechanisms and molecular networks underlying abdominal fat in chickens, in the present study, we used RNA-seq technology and integrated mRNA sequencing data to study the miRNA and mRNA expression profile of chicken abdominal fat preadipocytes on day 0 and day 10 of differentiation. The key negatively correlated miRNA–mRNA interaction networks and pathways associated with abdominal fat cell differentiation were identified. Our aim was to aid in understanding the molecular mechanisms of chicken abdominal fat deposition, and provide a basis for adipocyte differentiation of chickens.

## 2. Materials and Method

### 2.1. Preadipocyte Isolation, Differentiation, and Sample Collection

Preadipocytes from 14-day-old Gushi chicken abdominal adipose tissue were cultured in accordance with the method described by Zhang [1]. Abdominal adipose tissue was collected from eight 14-day-old Gushi chickens under sterile conditions. After reaching 90% cell confluence, the induction medium was used to replace the basic medium for 48 h. Induced differentiation medium contains 0.5 mM 3-isobutyl-1-methylxanthine, 1 µM dexamethasone, 10 μg/mL insulin, and 300 µM oleic acid (Solarbio, Beijing, China). Then, the maintenance medium was replaced and cultured for 48 h. Then, the differentiation medium was replaced with maintenance medium (10 μg/mL insulin and 300 µM oleic acid) and incubated for 48 h. The d 0 cells were used as the control group, and the cells collected at d 10 of differentiation were used as the experimental group. These groups were named Ab-Pre and Ab-Ad, respectively (Figure 1A). See Appendix A for the overall outline of our experiment.

### 2.2. Oil Red O Staining 

The cells to be tested were collected, washed with PBS, and fixed with 4% paraformaldehyde for 30 min. After washing with PBS, the culture dish was completely dried. The cells were incubated for 20 min at room temperature with Oil Red O, then immediately cleaned with PBS, and visualized with light microscopy. After microscopic observation, 1 mL of isopropanol (100%) was added. Ten minutes later, when the Oil Red O was completely dissolved, the OD value of each plate was read and recorded at the wavelength of 500 nm by the enzyme reader, and the cell differentiation was analyzed.

### 2.3. RNA Extraction, Small RNAs Library Construction, and Deep Sequencing

Total RNA of Ab-preadipocytes and Ab-adipocytes was prepared using Illumina^®^ Small RNA Sample Prep Kit (NEB, Ipswich, MA, USA). Using Agilent Bioanalyzer 2100 (Agilent Technologies, Santa Clara, CA, USA) to evaluate RNA purity, the threshold RNA integrity number was >8. The total RNA was stored at −80 °C until used. After the quality examination, the purified total RNA 3′ and 5′ were connected to the adapter, respectively, according to the protocol (T4 RNA Ligase 2 truncated, BioLabs, USA), and cDNA was obtained by reverse transcription–polymerase chain reaction (PCR). Then, the cDNA was amplified by PCR. The sequence library was constructed from the amplified products obtained from agarose gel. A total of 4 libraries were sequenced with Illumina Genome Analyzer (Illumina, San Diego, CA, USA). Four small RNA libraries were sequenced for adipocytes, which were designated Ab-Pre-1, Ab-Pre-2, Ab-Ad-1, and Ab-Ad-2.

### 2.4. Data Analyses

The raw reads were filtered using the Fastx pre-processing tool to remove the 3′ and 5′ junction sequences, low-quality reads, reads smaller than 18 nt or longer than 40 nt [17]. Bowtie (http://bowtie.cbcb.umd.edu) was applied to align the raw reads with the reference genome (version: Gallus gallus-5.0, serial number: GCA_000002315.3); the alignment process allows one base mismatch. Clean reads of each sample were compared with databases such as miRBase to annotate RNA. The miRCat tool in the sRNA Toolkit package was used to predict novel miRNAs [18,19].

### 2.5. Differential Expression Analyses

In order to compare the expression patterns of miRNAs between preadipocytes and differentiated adipocytes, we standardized the expression of miRNAs by TPM, which refers to the number of reads from a transcript per million reads [20]. Then, edgeR was used to analyze the differential expression of the data [21]. The Benjamin–Hochberg method was used for multiple hypothesis tests to control the false discovery rate. After correcting the *p*-value, the *p*-value was ≤0.05, which served as the threshold for differential expression. miRNAs with fold changes ≥2 or ≤0.5 were identified as differentially expressed miRNAs (DEMs). 

### 2.6. Prediction and Functional Analyses of miRNA Target Genes

To determine the biological function of target genes, we used the David database (https://DAVID.ncifcrf.gov/) to analyze the predicted target gene by gene ontology (GO) enrichment and Kyoto Encyclopedia of Genes and Genomes (KEGG). With *p*-value ≤ 0.05 as the threshold, only GO-terms and KEGG paths satisfying this condition were considered. 

### 2.7. Validation of miRNAs and mRNAs Expression by qRT-PCR

The quantitative real-time reverse transcription PCR (qRT-PCR) method was used to detect the relative expression levels of six DEMs and DEGs randomly selected. According to the manufacturer’s instructions, the total RNA was retrieved using the primescript RT kit (TaKaRa, Dalian, China), and chicken U6 RNA and β-actin were selected as the internal reference for miRNAs and mRNAs, respectively. It was carried out on a LightCycler 96 instrument (Roche, Indianapolis, IN, USA) with a final volume of 10 µL. The 2^−ΔΔCt^ method was used to calculate the relative expression of miRNAs and mRNAs. The loop primers used for the qRT-PCR were ordered from Shanghai GenePharma Co., Ltd. (Shanghai, China). Specific primers for genes were ordered from Henan Shangya Biotechnology Co., Ltd. (Appendix A). Differences between the Ab-Pre groups and Ab-Ad groups were analyzed by Student’s *t*-tests using GraphPad Prism 7 (GraphPad Software, version: GraphPad Prism 7.0, San Diego, CA, USA). All the reactions were run in three biological replicates and two technical replicates.

### 2.8. miRNAs–mRNAs Interaction Analysis

We used miRanda to predict the target genes of DEMs [22], and the cross part between the target genes predicted by DEMs and the DEGs (published) was defined as the intersection genes [23]. Using the intersection genes to carry out the analysis of KEGG enrichment, the pathways related to lipid metabolism in the KEGG enrichment pathway with *p*-value < 0.5 were visualized. In addition, in order to better understand the interaction between DEMs and the cross genes, Pearson correlation analysis was used to determine the negatively correlated miRNA–mRNA pairs, using Cytoscape software for the miRNAs–mRNAs network. We also constructed a hypothetical regulatory network to show the interaction of cross genes and DEMs in the process of lipid metabolism in chickens, which also included key miRNA and negatively correlated mRNA target networks. 

## 3. Results

### 3.1. Model Construction between Ab-Pre and Ab-Ad Group In Vitro

In order to detect the changes of lipid droplets of abdominal preadipocytes on 0d and 10d, the staining results of Oil Red O were shown in Figure 1B. Compared with 0d, the lipid droplets of cells cultured in differentiation medium for 10d were significantly increased. 

### 3.2. General Description of the Small RNA-seq Data 

A total of 24317079, 20450354, and 18449530, 20749337 raw reads were obtained from the Ab-Pre-1, Ab-Pre-2 and Ab-Ad-1, Ab-Ad-2 libraries, respectively (Table 1). After removing contaminant reads, we obtained 20332045, 19600041 (Ab-Pre) and 18133233,20329584 (Ab-Ad) clean reads in the range of 18–40 nt which were used for subsequent analyses. These clean reads were uniquely mapped to the chicken genome (Gallus gallus 5.0), and the average mapping clean reads of Ab-Pre and Ab-Ad libraries were 15453009 and 16688352, respectively. Most of the clean reads are 21–24 nt, of which 22 nt is the most abundant, followed by 23 nt. The sequenced sRNAs were mapped to many public databases, and to more than 70% of miRNAs (Appendix A). However, the proportion of mapped other RNAs such as rRNAs, tRNAs, mRNAs, and snoRNA was low. In addition, the proportion of base quality greater than Q20 is as high as 99%. Therefore, the samples in this study all meet the quality control indexes and can be used for the subsequent differential expression analysis.

### 3.3. Differential Expression Analysis of miRNAs between Ab-Pre and Ab-Ad Groups

To identify the DEMs that may play important regulatory roles in abdominal fat production, 1055 miRNAs were identified in Ab-Pre groups and Ab-Ad groups, and 80 miRNAs with different expression levels were screened. The differentially expressed 80 miRNAs (Appendix A), included 13 novel miRNAs, 58 of which were up-regulated, and 22 of which were down-regulated (Figure 2A). Five conserved families were all differentially expressed with *p*-value ≤ 0.05, including miR-130 (miR-130a, -130c), let-7 (let-7a, -7c), miR-18 (miR-18a, -18b), miR-29 (miR-29a, -29b, -29c), and miR-30 (miR-30a, -30c). All members of the five families were down-regulated in Ab-Ad compared with Ab-Pre. In 13 novel miRNAs, seven novel miRNAs were only expressed in the Ab-Pre group, three novel miRNAs were detected in only the Ab-Ad group, and three novel miRNAs were shared between the libraries. The top 10 highly expressed miRNAs were significantly expressed in both libraries (Figure 2B); miR-101-3p, miR-10a-5p, miR-146b-5p, miR-199-3p, and miR-214 were up-regulated in Ab-Ad compared with Ab-Pre, whereas the other miRNAs were up-regulated. miR-101-3p exhibited the highest expression levels in both libraries, followed by miR-10a-5p and miR-146b-5p, whereas miR-148a-5p (−2.54-fold) exhibited the maximum fold change (Additional File 4: Appendix A). Sample correlation, PCA, and cluster analysis showed that the two repetitive similarities of Ab-pre and Ab-Ad groups were good. The volcano plot showed miRNAs with different expression patterns (Appendix A).

### 3.4. qRT-PCR Validation of the Sequencing Data

We used qRT-PCR to verify the reliability of sequencing data results. Six miRNAs (miR-130a-5p, miR-18a-3p, miR-18b-3p, miR-223, miR-106-3p, and miR-27b-5p) and mRNAs (*ELOVL5*, *COL6A1*, *MAPK10*, *G6PC2*, *ROCK2*, and *BMP4*) with different expression levels were selected for analysis. The results showed that the expression profiles of these miRNAs and mRNAs had similar expression patterns with sequencing data (Figure 3).

### 3.5. Prediction of Target Genes of Differentially Expressed miRNAs and Functional Annotation

Of the DEMs between the Ab-Pre and Ab-Ad groups, 14,524 target genes were predicted. To identify the possible biological function of target genes, enrichment analysis of GO function and KEGG pathway was carried out. The results showed that the biological processes of enrichment of target genes included cell differentiation (GO: 0030154), regulation of cell differentiation (GO: 0045595), and steroid biosynthetic process (GO: 0006694). In addition, the results of KEGG enrichment were also associated with lipid metabolism and KEGG enrichment pathways including the PPAR signaling pathway, biosynthesis of unsaturated fatty acids, glycerophospholipid metabolism, insulin signaling pathway, and adipocytokine signaling pathway. Besides, the top 30 KEGG enrichment pathways also contains a lot of pathways related to fatty acid synthesis (Figure 4A) such as fatty acid metabolism, fatty acid elongation, steroid biosynthesis, peroxisome, and fatty acid degradation. The above pathways mainly include the ELOVL fatty acid elongase family (*ELOVL1*, *ELOVL4*, *ELOVL5*, and *ELOVL6*), fatty acid-binding protein family (*FABP3*, *FABP5*), perilipin family (*PLIN1*, *PLIN2*), acetyl-CoA acyltransferase family (*ACAA1*, *ACAA2*), acyl-CoA synthetase bubblegum family member (*ACSBG1*, *ACSBG2*), and stearoyl-CoA desaturase family (*SCD*, *SCD5*), *EHHADH*, *FASN*, *LPL*, *HSD17B12*, and *ACOX1*.

### 3.6. Integrated Analysis of DEMs and DEGs Involved in Lipid Metabolism

A total of 4693 DEGs were obtained in the Ab-Pre groups and Ab-Ad groups, 2797 of which were significantly up-regulated and 1896 significantly down-regulated. We analyzed the DEGs and target genes predicted by DEMs, and identified 891 cross genes (Appendix A). In addition, we analyzed the function enrichment of 891 genes (Appendix A). These genes were annotated to the total of 117 signaling pathways, such as nine genes annotated to ECM–receptor interaction and 15 genes annotated to the MAPK signaling pathway, and other classic cellular signaling pathways. Six genes were annotated to the adipocytokine signaling pathway. Twelve pathways related to lipid metabolism were enriched (Figure 4B). Based on the above enrichment pathways, a hypothetical regulatory network shows the interaction between DEGs and DEMs (Figure 5). In this network, except for the citrate cycle (TCA cycle), all other signaling pathways are connected. 

In addition, there are many genes in the MAPK signaling pathway, such as *FOS*, *DUSP6*, *CACNA1G*, and *MAPKAPK3*. Interestingly, *MAPK10* is part of the MAPK signaling pathway, insulin signaling pathway, fatty acid metabolism, foxo signaling pathway, and was targeted by nine miRNAs, including miR-29b-3p, miR-6670-5p, miR-135a-5p, miR-1625-5p, miR-1604, miR-1774, miR-1737, miR-6675-3p, and miR-301a-5p. In the insulin signaling pathway, *SOCS3* and *G6PC2* are important genes. *SOCS3* interacts with the adipocytokine signaling pathway, and *SOCS3* is regulated by miR-1703-5p. *G6PC2* interacts with the foxo signaling pathway and glycolysis/gluconeogenesis. *G6PC2* is targeted by miR-29b-3p. *ACSBG2* is the bridge of communication between the adipocytokine signaling pathway and fatty acid metabolism, and *ACSBG2* is the target of miR-214. ECM–receptor interaction is closely related to focal adhesion and regulation of actin cytoskeleton. *LAMA4* is targeted by miR-146b-5p and miR-6615-5p, *CHAD* is targeted by miR-1712-3p and miR-1650, and *SSH2* and *FLT4* are targeted by nine and seven miRNAs, respectively. In the above pathways, we obtained a total of 73 combined pairs of miRNAs–mRNAs, including 10 pairs with high expression of miRNA regulation, such as seven genes regulated by miR-214 (*ACSBG2*, *ITGB2*, *PLK3*, *CCND3*, *ACLY*, *CAMK2A*, and *NFKB2*), miR-148a-5p, miR-10a-5p, and miR-146b-5p negatively regulate *ROCK2*, *ELOVL5*, and *LAMA4*, respectively. Among the 73 combined pairs, two pairs were associated with the high expression of top 20 mRNAs, which were miR-6615-5p regulating *FLNB* and miR-1774 regulating *COL6A1*, respectively (Figure 6). 

## 4. Discussion

To date, most studies of the chicken transcriptome have focused on the tissue level, but few have focused on the cell-level. Most cell-level studies have focused on growth. The screening and identification of key genes for fat deposition in pigs by cell transcriptome sequencing technology has become relatively mature in China [24,25]. This study is the first time to construct the differentiation model of Gushi chicken abdominal fat cells in vitro and screen the key miRNAs and mRNAs of chicken fat deposition. 

Among the known miRNAs identified, we found that the expression of miRNAs is consistent with the previous results, such as miR-146b, miR-130a, miR-27b-5p, and miR-30a. Studies have shown that overexpression of miR-146b promotes 3T3-L1 cell differentiation [26] and human visceral adipocyte differentiation [27], overexpression miR-130a in porcine preadipocytes can inhibit its differentiation [28], and the depletion of miR-27b promotes lipid accumulation and weight gain in zebrafish [29]. In addition, most abundantly expressed miRNAs, such as miR-30a and miR-106, have been reported that inhibition of miR-30a can promote adipocyte differentiation [30] and miR-106 as negative regulators of adipocyte differentiation [31]. In this study, among the miRNAs, the top 10 abundantly expressed miRNAs have not been studied, such as miR-101-3p, miR-10a-5p, miR-1559-5p, and miR-458a-3p. On the other hand, in addition to the known differences in miRNAs, it is also necessary to study the functions of novel miRNAs, although identifying the functions of these novel miRNAs may be challenging.

qRT-PCR has the advantages of high sensitivity, specificity, and small error. At present, the technology has been widely used in animal, plant and human gene diagnosis, disease diagnosis, gene expression, and microorganisms and other fields. It is also a recognized method to verify the functions of miRNAs and target genes. In this study, six DEMs and DEGs were selected from the two libraries for qRT-PCR validation, and the results were consistent with the sequencing results. However, in order to verify the authenticity of high-throughput sequencing, in principle, all detected DEMs and DEGs should be validated rather than just randomly selecting six of them to verify the accuracy of the results, which is also the limitation of this experiment.

At the cellular level, fat deposition is the result of an increase in the number of fat cells and the volume of individual cells. Among them, the number of adipocytes is determined by the degree of differentiation of pluripotent stem cells into preadipocytes, and the volume of a single cell is related to its differentiation degree and triglyceride accumulation. Therefore, the differentiation degree of adipocytes can explain the problem of fat deposition. However, adipocyte differentiation is a complex physiological process and requires a series of combinatorial biological processes to determine whether to differentiate [32]. In this study, the pathway of cross genes enrichment is related to adipocyte differentiation, including regulation of actin cytoskeleton, the MAPK signaling pathway, citrate cycle (TCA cycle), ECM–receptor interaction, focal adhesion, foxo signaling pathway, and insulin signaling pathway. It has been reported that during the process of adipocyte differentiation, cell morphology will change. With the change of cell morphology, the synthesis and assembly of actin and other major cytoskeleton proteins will also change [33], so the pathways involved in cell growth, proliferation, migration, and differentiation may affect lipid deposition. In fact, classical cell signaling pathways such as the MAPK signaling pathway and citrate cycle (TCA cycle) play an indispensable role in many cell responses, such as cell proliferation and differentiation [34,35]. In addition, ECM is an important component of cell microenvironment [36], which actively participates in signal transduction of cell behavior and provides physical support for cells [37,38]. A study has shown that the extracellular environment of intramuscular and subcutaneous preadipocyte may play an important role in adipocyte differentiation [39]. Focal adhesions act as a cellular regulator of cell growth, movement, and differentiation [40,41]. FOXOs are implicated in a broad range of cellular functions, and these are cellular properties, critical to the cell biology, including cellular differentiation, apoptosis, cell proliferation, DNA damage, and repair [42,43]. Obesity leads to the progression of insulin resistance and type II diabetes mellitus [44], and studies have reported that insulin signaling pathways regulate adiposity cell differentiation [45]. Therefore, these signaling pathways may be keys to further explore the molecular mechanisms underlying chicken adipose differentiation. 

Combining target genes of DEMs with differentially expressed mRNA data can improve the accuracy of target gene prediction. Compared with Huang’s research, many of our intersection genes are the same, such as *SCD*, *APOA1*, *LAMA2*, *ROCK2*, and *CHAD*. In addition, the enriched signal pathways such as fatty acid metabolism, ECM–receptor interaction, focal adhesion, peroxisome pathway are also the same [46]. Of course, there are different genes and enriched signal pathways. We speculate that the possible cause is the difference in breeds and age of samples. In addition, Huang’s research is conducted at the individual level, while our research is conducted at the cell level. 

Furthermore, in our study, most of the intersection genes have been reported to be related to adipocyte differentiation, such as *ACSBG2*, *SOCS3*, *PAK1*, *MAPK10*, and *G6PC2*. ACSBG enzyme, also as known as lipase, has been identified as two members of the ACSBG gene family, *ACSBG1* and *ACSBG2* [47], which mainly activate fatty acids with C16 to C24 [48,49]. *SOCS3* is a key regulator of the JAK-STAT pathway, [50] and studies have shown that *SOCS3* plays a key role in leptin and insulin resistance [51,52]. *SOCS3*-specific knockout mice have shown improved diet-induced obesity, leptin, and insulin resistance compared with wild-type mice [53]. This suggests that *SOCS3* plays an important role in lipid accumulation. Cell motility requires polarized rearrangements of the actin/myosin cytoskeleton, and PAKs affect a series of processes that are crucial to the cell from the regulation of cytoskeletal remodeling, cell motility, morphology, and cell proliferation [54]. As a member of the PAK family, *PAK1* has been shown to be involved in extracellular signaling that modulates cell polarity and actin cytoskeleton organization [55,56]. Studies have shown that inducing the expression of wild-type *PAK1*, kinase-dead, or constitutively-active forms of this enzyme *PAK1* in NIH-3T3 cells can cause significant changes in actin tissue [57]. However, the morphology of preadipocytes changes during differentiation. JNKs play a pivotal role in adipocyte differentiation, obesity, and insulin resistance [58]. As a member of the JNKs family, it has been reported that *MAPK10* participates in many cellular processes, such as proliferation, differentiation, transcriptional regulation, and development [59]. In addition, *MAPK10* is related to fat [60], liver development [61], muscle atrophy [62], type I diabetes, and muscle-invasive tumors [63]. It has been reported that glycerol can induce glucose-6-phosphatase, the rate-limiting enzyme of gluconeogenesis [64], while glucose-6-phosphate must be acted upon by *G6PC2* to become glucose, and then the glucose produced can be utilized by muscle and fat and other tissues [65]. 

It has been argued that a single miRNA can simultaneously regulate the expression of multiple genes, and one gene can be regulated by multiple miRNAs [66]. miR-214 has been previously implicated in fibroblast differentiation of adipose-derived mesenchymal stem cells by targeting *MFN2* [67], and regulates the differentiation of human hair follicle stem cells and bone marrow-derived mesenchymal stem cells [68,69]. *ACLY*, the putative target gene of miR-214, connects glucose metabolism to de novo synthesis of lipids [70]. *ITGB2* is also associated with fat [71]. MiR-148a is associated with adipocyte differentiation [72]. *ROCK2*, the target gene of miR-148a-5p, is an anti-lipogenic factor [73], furthermore, *ROCK2* plays an important role in the remodeling of actin cytoskeleton, and the loss of actin fibers is necessary for adipogenesis [74]. The target gene *LAMA4* of miR-146b-5p and the high expression of *COL6A1* are related to focal adhesion and ECM–receptor interaction. It has been reported that in the process of adipocyte differentiation in vivo, changes in extracellular matrix composition, and organization can affect the remodeling of cell morphology and cytoskeleton structure. This biological process is generally controlled by related signaling and adhesion [75]. *FLNB* is one of the members of the filamin family, which regulates many processes including cell motility and organ development [75]. Studies above suggested that these miRNAs and mRNAs might act as a crucial factor in adipocyte differentiation, so they can be used as subjects for further study in this research. This study not only helps to understand and clarify the regulation mechanism of chicken abdominal fat deposition, but also to lay a foundation for genetic breeding and nutrition regulation of fat deposition of chickens. 

## 5. Conclusions

In conclusion, the DEMs, DEGs, and pathways we found may play an important role in the differentiation of abdominal preadipocyte, and lay a foundation for the study of the molecular mechanism of fat deposition. In the future, we will further determine the potential molecular mechanism between some important candidates and abdominal fat deposition, and provide some valuable resources for the subsequent breeding of low abdominal fat chickens.

## Figures and Tables

**Figure 1 animals-10-00468-f001:**
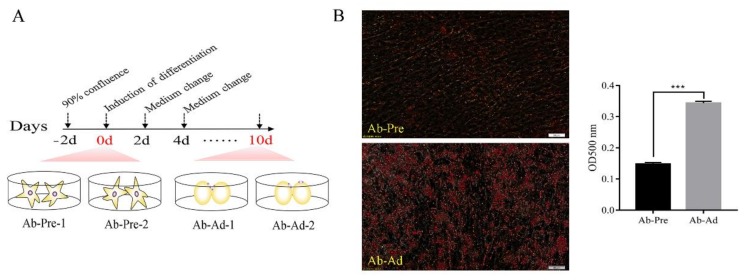
General description of the small RNA-seq data. (**A**) Procedure for inducing the differentiation of abdominal cells. Cells were collected for RNA-Seq at day 0 (Ab-Pre) and day 10 (Ab-Ad). Each stage included two biological replicates. (**B**) Oil Red O staining of preadipocytes (0d) and adipocytes (10d). The data are demonstrated as the means ± SEM; *** means *p* < 0.001.

**Figure 2 animals-10-00468-f002:**
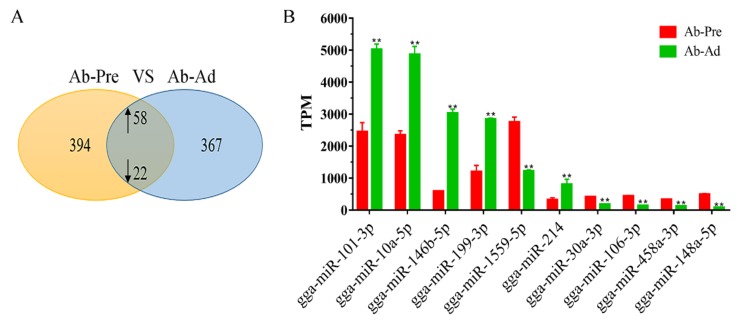
Analysis of differentially expressed miRNAs. (**A**) The Venn diagrams of the differentially expressed miRNAs (DEMs) between the preadipocytes (Ab-Pre) and differentiated adipocytes (Ab-Ad) groups. (**B**) Top 10 most abundantly expressed miRNAs in the preadipocytes (Ab-Pre) and differentiated adipocytes (Ab-Ad). The data are demonstrated as the means ± SEM; ** means *p* < 0.01.

**Figure 3 animals-10-00468-f003:**
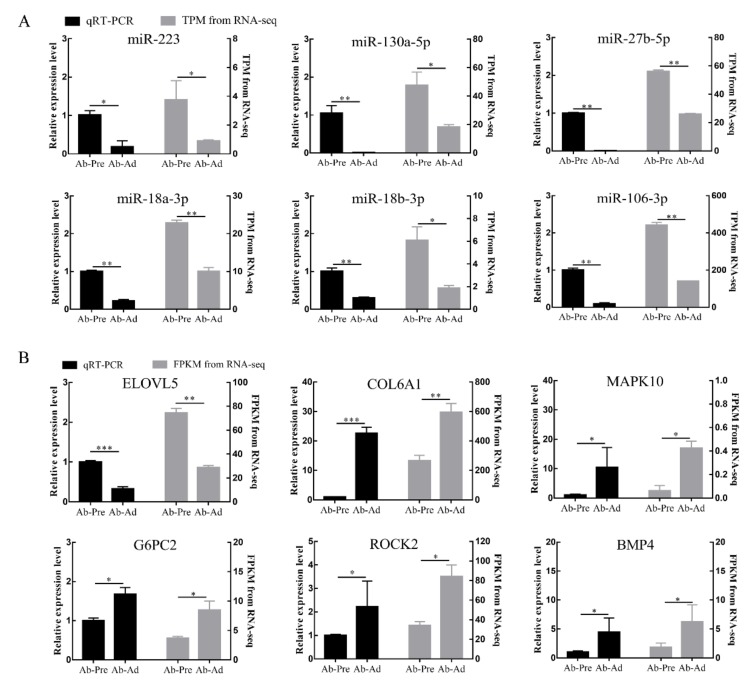
Verification of the accuracy of sequencing results by qRT-PCR. (**A**) Validation results of miRNAs. (**B**) Validation results of mRNAs. The data are demonstrated as the means ± SEM; * means *p* < 0.05, ** means *p* < 0.01, *** means *p* < 0.001.

**Figure 4 animals-10-00468-f004:**
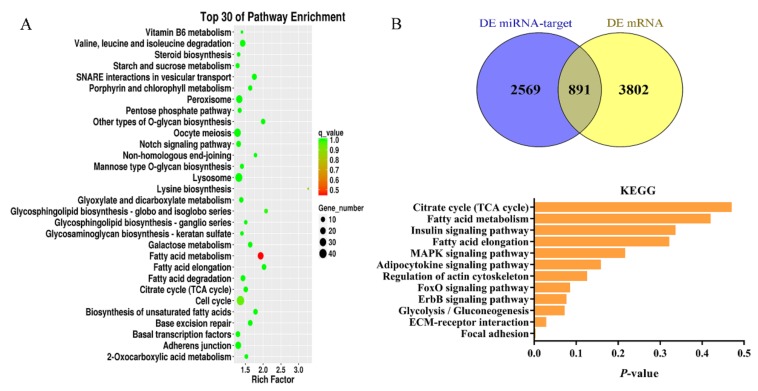
**(A**) Kyoto Encyclopedia of Genes and Genomes (KEGG) enrichment analysis reveals the top 30 enrichment pathways. (**B**) The Venn diagrams of the DEMs target genes and DEGs between the preadipocytes (Ab-Pre) and differentiated adipocytes (Ab-Ad) groups, KEGG enrichment analysis involved in lipid metabolism.

**Figure 5 animals-10-00468-f005:**
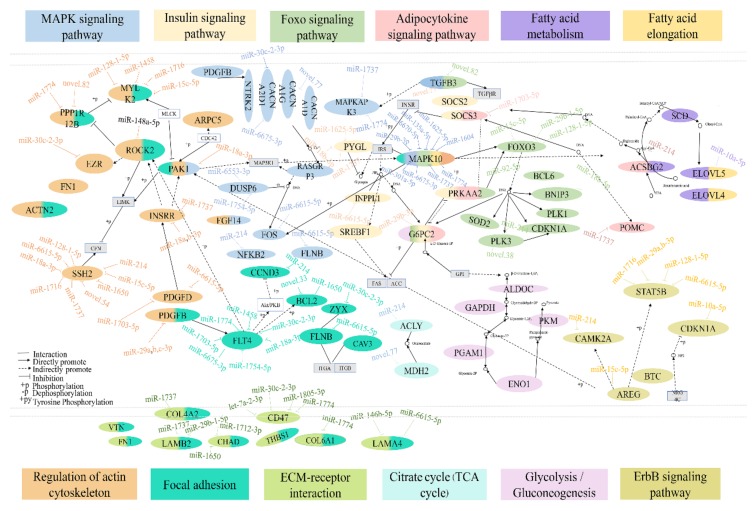
An interaction network of the DEMs and DEGs across twelve pathways during lipid metabolism in chickens.

**Figure 6 animals-10-00468-f006:**
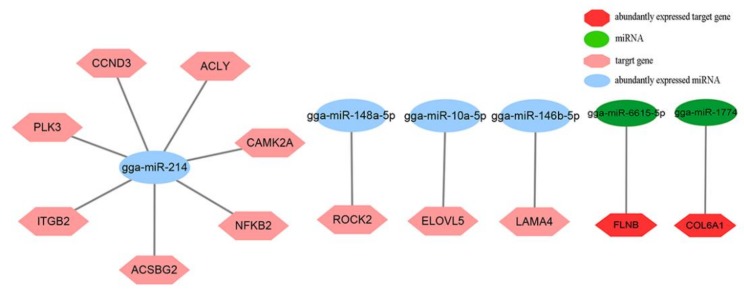
Integrated analysis of the abundantly expressed miRNAs–mRNAs networks and abundantly expressed miRNAs–abundantly mRNAs.

**Table 1 animals-10-00468-t001:** Overview of sequencing data for small RNA.

Samples	Raw Reads	Clean Reads	Clean Ratio (%)	Mapped Reads	Mapped Ratio (%)	Q20 Value
Ab-Pre-1	24317079	20332045	83.61%	14782763	72.3%	99.40%
Ab-Pre-2	20450354	19600041	95.84%	16123255	82.3%	99.54%
Ab-Ad-1	18449530	18133233	98.29%	15667237	86.4%	99.24%
Ab-Ad-2	20749337	20329584	97.98%	17709466	87.1%	99.59%

Note: Ab-Pre—samples from 0d undifferentiated preadipocytes; Ab-Ad—samples from preadipocytes after 10d differentiation. Clean ratio—clean reads/raw reads. Mapped ratio—reads that matched the reference genome completely. The proportion of base quality greater than Q20 is not less than 90%. Q20 Value= bases of quality greater than or equal to 20/all bases of sequencing.

## Data Availability

All the Illumina miRNA-seq data sets supporting the results of this article have been submitted to the National Center for Biotechnology Information (NCBI) Gene Expression Omnibus (GEO) under accession number SRR7067303, SRR7067304, SRR7067305, SRR7067306.

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
