# Peer review of "MiRNAs and mRNAs Analysis during Abdominal Preadipocyte Differentiation in Chickens"

_animals, 2020, doi:10.3390/ani10030468_

Round 1
Reviewer 1 Report
Dear authors,
I no longer have anything to revise.
Author Response
Thank you for your comments.Reviewer 2 Report
This is an interesting paper, very well performed and redacted. English is correct and very easy to follow. In general, the authors must present the Gushi breed as an animal model to extend the findings to the whole meet chickens. I recommend, as well, to reduce the supplementary information because is excessive and some not relevant.
I recommend the acceptance of this paper after the following amendments:
Line 17.- Remove ”and play a positive role in improving chicken production efficiency”. It is not demonstrated, or at least put it in conditional.
Lines 67-69.- Please mention the reference of “ It was reported that the heritability of 68 abdominal fat (0.82) was significantly higher than that of live weight (0.55), breast muscle (0.55), leg 69 (0.51) and thigh (0.31) at the time of selection”. These data are excessive, and I suspect a not valuable research.
Line 70.- Remove Gushi. It affects to all chickens.
Line 77.- Remove “and breed selection of Gushi chicken”, it is rather ambitious.
Line 81.- “2W”, what´s that?, please describe it for not experts
Line 145.- Replace “pairs. Using” by “pairs, using”
Lines 153-154.- “indicating that 154 triacylglycerol content in the Ab-Ad was significantly higher than that in Ab-Pre.” It belongs to the discussion section.
Line 170.- “subsequent data analysis”, which one?
Lines 205-208.- Figure 5 is not relevant, it could go to additional material
Lines 256-258.- “Above all, these results revealed that the above 12 pathways and these 12 DEGs–DEMs interactions play critical roles in chicken abdominal fat metabolism”. It belongs to the discussion section.
Lines 259-262.- Improve figure 5. It is not recommendable to present figures with information impossible to read. If it is not possible, please remove.
Line 273.- “in vitro” all Latin terms must be presents in italics please review all the manuscript.
Lines 295-312.- These contents are more a review than a discussion. The own results must be related more deeply to this information. Please resolve it.
Line 345.- Replace “MFN2 [72]. and regulates” by “MFN2 [72], and regulates”
Line 352.- “in vivo” Latin term, please present it in italic
Line 358.- Replace “in this study” by “in this research”
Lines 361-367.- Conclusions must be completely reformulated. Presently are a description of what the authors have done and some speculations. Authors must present here the projection of the findings both in the science and in the productive sector. For instance, is they predict a use in breeding must be remarked how it could be.
Author Response
This is an interesting paper, very well performed and redacted. English is correct and very easy to follow. In general, the authors must present the Gushi breed as an animal model to extend the findings to the whole meet chickens. I recommend, as well, to reduce the supplementary information because is excessive and some not relevant.
Answer: Thank you very much for your valuable advice. Our supplementary information is related to the article, if deleted, it will affect the integrity of the article.
I recommend the acceptance of this paper after the following amendments:
Line 17.- Remove ”and play a positive role in improving chicken production efficiency”. It is not demonstrated, or at least put it in conditional.
Answer: Thank you very much for your advice. Sorry for the inappropriate description. We have deleted this sentence.
Lines 67-69.- Please mention the reference of “ It was reported that the heritability of 68 abdominal fat (0.82) was significantly higher than that of live weight (0.55), breast muscle (0.55), leg 69 (0.51) and thigh (0.31) at the time of selection”. These data are excessive, and I suspect a not valuable research.
Answer: Thank you very much for your advice. We have inserted the references.
Line 70.- Remove Gushi. It affects to all chickens.
Answer: Thank you very much for your advice. Sorry for the inappropriate description. We have deleted this sentence.
Line 77.- Remove “and breed selection of Gushi chicken”, it is rather ambitious.
Answer: Thank you very much for your advice. Sorry for the inappropriate description. We have deleted this sentence.
Line 81.- “2W”, what´s that?, please describe it for not experts
Answer: Thank you very much for your advice. Sorry for the inappropriate description. We have changed it to 14-day-old.
Line 145.- Replace “pairs. Using” by “pairs, using”
Answer: Thank you very much for your advice. We have revised it in the latest version.
Lines 153-154.- “indicating that 154 triacylglycerol content in the Ab-Ad was significantly higher than that in Ab-Pre.” It belongs to the discussion section.
Answer: Thank you very much for your advice. Sorry for the inappropriate description. We have deleted this sentence.
Line 170.- “subsequent data analysis”, which one?
Answer: Thank you very much for your advice. Sorry for the inappropriate description. We have changed: Therefore, the samples in this study all meet the quality control indexes and can be used for the subsequent differential expression analysis. (line 171-172)
Lines 256-258.- “Above all, these results revealed that the above 12 pathways and these 12 DEGs–DEMs interactions play critical roles in chicken abdominal fat metabolism”. It belongs to the discussion section.
Answer: Thank you very much for your advice. Sorry for the inappropriate description. We have deleted this sentence.
Lines 259-262.- Improve figure 5. It is not recommendable to present figures with information impossible to read. If it is not possible, please remove.
Lines 205-208.- Figure 5 is not relevant, it could go to additional material
Answer: Thank you very much for your advice. We have changed the text of the figures, and hope it can meet the requirements. Line205-208 is figure 3.
Line 273.- “in vitro” all Latin terms must be presents in italics please review all the manuscript.
Answer: Thank you very much for your advice. We have revised it in the latest version.
Lines 295-312.- These contents are more a review than a discussion. The own results must be related more deeply to this information. Please resolve it.
Answer: Thank you very much for your advice. We have revised it in the latest version. (line 300-303)
Line 345.- Replace “MFN2 [72]. and regulates” by “MFN2 [72], and regulates”
Answer: Thank you very much for your advice. We have revised it in the latest version.
Line 352.- “in vivo” Latin term, please present it in italic
Answer: Thank you very much for your advice. We have revised it in the latest version.
Line 358.- Replace “in this study” by “in this research”
Answer: Thank you very much for your advice. We have revised it in the latest version.
Lines 361-367.- Conclusions must be completely reformulated. Presently are a description of what the authors have done and some speculations. Authors must present here the projection of the findings both in the science and in the productive sector. For instance, is they predict a use in breeding must be remarked how it could be.
Answer: Thank you very much for your valuable advice. Sorry for the inappropriate description. We have changed the conclusions as follows: In conclusion, the DEMs, DEGs and pathways we found may play an important role in the differentiation of abdominal preadipocyte, and lay a foundation for the study of the molecular mechanism of fat deposition. In the future, we will further determine the potential molecular mechanism between some important candidates and abdominal fat deposition, and provide some valuable resources for the subsequent breeding of low abdominal fat chickens. (line 369-373)
Reviewer 3 Report
Please provide references for the descriptions:
L44: Our previous studies have shown that abdominal preadipocytes have higher adipogenic differentiation ability than intramuscular preadipocytes, so reducing abdominal fat deposition is of great significance to poultry industry.
L67: It was reported that the heritability of abdominal fat (0.82) was significantly higher than that of live weight (0.55), breast muscle (0.55), leg (0.51) and thigh (0.31) at the time of selection.
1.In my understanding, adipocyte differentiation and fat deposition are different concepts. The main objective of this study is to find the molecular mechanism of abdominal fat deposition in chicken. But it seems that the results were limited to the key genes involved in adipocyte differentiation. How can readers interpret the results?
Answer: Thank you very much for your advice. Sorry for the unclear description. At the cellular level, fat deposition is the result of the increase of the number of fat cells and the volume of single cells. Among them, the number of adipocytes is determined by the degree of differentiation of pluripotent stem cells into preadipocytes, while the volume of single cell is related to its differentiation degree and triglyceride accumulation, so adipocyte differentiation can explain the problem of fat deposition from the side.
Please add descriptions about the above answers in the section of the discussion. This point should be clarified.
- How many biological replicates and technical replicates are there? The preadipocytes were cultured from ONLY one animal and ONLY two samples per treatment were sequenced. Is it enough to have statistical significance and meaning from the differential expression analysis?
Answer: Thank you very much for your advice. In the present study, we isolated preadipocytes from 8 Gushi chickens, which have been changed in the material method part of this article (line 132). two biological replicates in each group were used for RNA-seq. All RT-PCR tests were performed in three biological replicates and two technical replicates. Pearson correlation analysis and Principal component analysis (PCA) showed global differences among the preadipocyte and adipocyte (Supplementary Figure S2). All evidence suggested that our data was repeatability and reproducibility. In addition, previous studies suggested that although this might hide the individual variation, considering the limited research funding of our research and the high repeatability of cell samples, two biological replicates for each condition was selected. It can satisfy the difference
I am sorry but it still unclear to me. I can’t find the added description on Line 132 in the revised manuscript. Also, why the numbers of chickens for RNA-seq and RT-PCR? Were RNA-seq and RT-PCR conducted from different chickens? In addition, 3 technical replicates per biological replicate are used for qRT-PCR, generally.
- How many miRNAs were evaluated for the differential expression analysis? How p-values were corrected?
Answer: Thank you very much for your advice. We have 1055 miRNAs for differential expression analysis. We have changed in the results section. (line 251-253)
Sorry for the unclear description. We have changed in the material and method section: After p-value is corrected by multiple hypothesis test, it is q-value ≤ 0.05, which served as the threshold of different expressions. (line 175)
Which method used for multiple hypothesis test? It should be provided.
Author Response
Please provide references for the descriptions:
L44: Our previous studies have shown that abdominal preadipocytes have higher adipogenic differentiation ability than intramuscular preadipocytes, so reducing abdominal fat deposition is of great significance to poultry industry.
L67: It was reported that the heritability of abdominal fat (0.82) was significantly higher than that of live weight (0.55), breast muscle (0.55), leg (0.51) and thigh (0.31) at the time of selection.
Answer: Thank you very much for your advice. We have inserted the references.
1.In my understanding, adipocyte differentiation and fat deposition are different concepts. The main objective of this study is to find the molecular mechanism of abdominal fat deposition in chicken. But it seems that the results were limited to the key genes involved in adipocyte differentiation. How can readers interpret the results?
Answer: Thank you very much for your advice. Sorry for the unclear description. At the cellular level, fat deposition is the result of the increase of the number of fat cells and the volume of single cells. Among them, the number of adipocytes is determined by the degree of differentiation of pluripotent stem cells into preadipocytes, while the volume of single cell is related to its differentiation degree and triglyceride accumulation, so adipocyte differentiation can explain the problem of fat deposition from the side.
Please add descriptions about the above answers in the section of the discussion. This point should be clarified.
Answer: Thank you very much for your valuable advice. We have added to the discussion section. (line 294-298)
- How many biological replicates and technical replicates are there? The preadipocytes were cultured from ONLY one animal and ONLY two samples per treatment were sequenced. Is it enough to have statistical significance and meaning from the differential expression analysis?
Answer: Thank you very much for your advice. In the present study, we isolated preadipocytes from 8 Gushi chickens, which have been changed in the material method part of this article (line 132). two biological replicates in each group were used for RNA-seq. All RT-PCR tests were performed in three biological replicates and two technical replicates. Pearson correlation analysis and Principal component analysis (PCA) showed global differences among the preadipocyte and adipocyte (Supplementary Figure S2). All evidence suggested that our data was repeatability and reproducibility. In addition, previous studies suggested that although this might hide the individual variation, considering the limited research funding of our research and the high repeatability of cell samples, two biological replicates for each condition was selected. It can satisfy the difference
I am sorry but it still unclear to me. I can’t find the added description on Line 132 in the revised manuscript. Also, why the numbers of chickens for RNA-seq and RT-PCR? Were RNA-seq and RT-PCR conducted from different chickens? In addition, 3 technical replicates per biological replicate are used for qRT-PCR, generally.
Answer: Thank you very much for your valuable advice. I'm sorry we're clearly marked. In the present study, we isolated preadipocytes from 8 Gushi chickens, which have been changed in the material method part of this article (line 82-83). All RT-PCR tests were performed in three biological replicates and two technical replicates (line 139-140).
RNA-seq and RT-PCR were randomly selected from the same sample of abdominal adipocytes.
In order to meet the number of cells needed for the experiment, eight chickens were selected to separate abdominal fat cells.
Your suggestion is right, 3 technical replicates per biological replicate are used for qRT-PCR, generally. In this study, although we are two technical repeats, we choose two technical repeats whose error is less than 0.5. The variation of biological repetition is much greater than that of technical repetition. It has been reported that the coefficient of variation of single PCR reaction is 0.92%. There are also three biological repeats and two technical repeats in the literature, so three biological repeats and two technical repeats can also meet the requirements.
Reference:
Arikawa E , Pan H , Sun Y , et al. P53-M Real-Time PCR Array for Multi-Gene Expression Profiling and Microarray Data Validation: RT2 Profiler PCR Array[J]. Journal of biomolecular techniques: JBT, 2007, 18(1):18.
- Nikula, A. West, M. Katajamaa, etc. A human ImmunoChip cDNA microarray provides a comprehensive tool to study immune responses[J]. 303(1-2):122-134.
Takahashi, Naoki, Lammens, Tim, Boudolf, Véronique, etc. The DNA replication checkpoint aids survival of plants deficient in the novel replisome factor ETG1[J]. Embo Journal, 27(13):1840-1851.
- How many miRNAs were evaluated for the differential expression analysis? How p-values were corrected?
Answer: Thank you very much for your advice. We have 1055 miRNAs for differential expression analysis. We have changed in the results section. (line 251-253)
Sorry for the unclear description. We have changed in the material and method section: After p-value is corrected by multiple hypothesis test, it is q-value ≤ 0.05, which served as the threshold of different expressions. (line 175)
Which method used for multiple hypothesis test? It should be provided.
Answer: Thank you very much for your valuable advice. Sorry for the inappropriate description. We have changed in the material and method section: Using Benjamin-Hochberg method to multiple hypothesis test for controlling the false discovery rate, after correcting p-value, it is q-value ≤ 0.05, which served as the threshold of different expressions. (line 120-122)
This manuscript is a resubmission of an earlier submission. The following is a list of the peer review reports and author responses from that submission.
Round 1
Reviewer 1 Report
The authors investigated miRNA expression patterns related to abdominal preadipocyte differentiation in chickens. The results were interesting. There are some comments and questions below.
1.
In my understanding, adipocyte differentiation and fat deposition are different concepts. The main objective of this study is to find the molecular mechanism of abdominal fat deposition in chicken. But it seems that the results were limited to the key genes involved in adipocyte differentiation. How can readers interpret the results?
2.
How many biological replicates and technical replicates are there? The preadipocytes were cultured from ONLY one animal and ONLY two samples per treatment were sequenced. Is it enough to have statistical significance and meaning from the differential expression analysis?
3.
How many miRNAs were evaluated for the differential expression analysis? How p-values were corrected?
4.
Although the title refers to the integrated analysis of miRNA and mRNA, there is no description and results about DEGs by RNA-seq from the adipocyte cells.
Other minor comments:
L49: Did this study target the post-transcription? Why did the authors mention it in this sentence?
L61: please remove “.” in “~ [15]. and ~”
L61: Promoting >>> promotes
L92: Information on RNA quality should be provided.
L123: superscript for -∆∆Ct
L132: Is it “p-value < 0.05”?
L163: Was there a quality issue in the AB-Pre-1 sample?
Reviewer 2 Report
Title is too long. It must be reduced to “miRNAs and mRNAs analysis during abdominal preadipocyte differentiation in chickens”
Summary and abstract are correct
Key words. I recommend that the key words must not be repeated in the title. It reduces the visibility of paper. Use for example: RNA sequencing, fat cells, avian production, …
Introduction:
Lines 42-47. This is not the truth, really this expression is surely linked to other productive traits, the improvement of this character must be observed considering the quantitative genetics and not directly. Please do this affirmation more smoothly.
Line 61. In “ [15]. and miR-26a Promoting” must be “[15] and miR-26a promoting
Material and methods:
Sample size (animals, cultures, etc) must be better defined
Line 94. Replace “Then” for “then”
Statistical associations between mi-RNA-mRNA in respect the fat deposition expressions is not well described
Results:
In general, the results are rather long, most of the contents are really discussion, sometimes they are repeated. I recommend a reconsideration of both sections by part of the authors in order to improve the following up of the text.
Size of the text in figures 1, 2 and 5 must be bigger, presently is not readable. It is possible increase the size of the text maintaining the size of the figures.
Discussion:
Discussion section is rather long, and some contents are repeated in the results, I newly recommend a reconsideration of both sections. It must be reduced and concreted because there is not a proportionality between the amount and relevance of the findings with respect the words used to present and discuss them.
Conclusions:
Conclusions must be reformulated. Presently they are a repetition of the results description and a proposal of the paper contents relevance for future approaches. Really, considering the present conclusions the paper publication is not justified. Authors must stand out the actual repercussion of the paper on Science and the new supplies for the society if they exist.
Supplementary material is excessive, they must be shortened
Reviewer 3 Report
Dear Authors,
The author's findings provide resources for studies based on miRNAs and mRNAs analysis to elucidate the mechanisms of lipid metabolism in chickens, and will also help to better understand chicken abdominal fat deposition.
However, I think that it is necessary to strengthen the reliability of the result by adding as much information as possible.
Thereupon, there are some questions and concerns about the manuscript that you might consider. Major comments:
Introduction:
1. Please clarify the purpose of this study.
Is the integrated analysis of differentially expressed miRNAs and mRNAs?
Searching (clarify) for a candidate gene during abdominal preadipocyte differentiation in chickens?
Also, please change the title accordingly. 2. Please describe the reason (usefulness) you have to do this in the research introduction. 3. If your group has any related research to this research, please introduce it in the introduction. Methods:
4. Please add sample information with reference to the following paper.
Genome-Wide Analysis of lncRNA and mRNA Expression During Differentiation of Abdominal Preadipocytes in the Chicken
Tao Zhang, Xiangqian Zhang, Kunpeng Han, Genxi Zhang, Jinyu Wang, Kaizhou Xie and Qian Xue, G3: Genes, Genomes, Genetics March 1, 2017 vol. 7 no. 3 953-966; https://doi.org/10.1534/g3.116.037069 5. Check the notation of statistical analysis (Methods or Figures). 6. All figures were not clear.
Therefore, I cannot review your manuscript.
I recommend that you put only the main figures in the main text.
Then, move the others to the Supplemental section.
7. There are several results sections.
It seems to state the conclusion, but the conclusion is not accurately stated.
Also, some sentences need to move to the methodology section.
Write a solid conclusion by including a figure or table.
In particular, there is no conclusion in Figure 3. 8. Since there is no coordination or conclusion, the discussion does not clarify or have any discussion.
Therefore, please revise the discussion so that it is easy to understand. 9. The research itself is very interesting, so I think it would be helpful to write a simple and easy-to-understand figure flow and discussion. 10. Please describe your future prospects at the end of the discussion.
11. Are there any similar studies done in other chickens?
If so, the results need to be compared in the discussion section.
In particular, comparison of gene and pathway data is necessary.
Integrated analysis of microRNA and mRNA expression profiles in abdominal adipose tissues in chickens, HY Huang, RR Liu, GP Zhao, QH Li, MQ Zheng… - Scientific reports, 2015 - nature.com. Article number: 16132 (2015)
12. In Acknowledgments section
Authors wrote “I would like to….” in the section.
Did all authors confirm your manuscript?
13. Please revise some English texts.
